# Nanomaterials Application in Endodontics

**DOI:** 10.3390/ma14185296

**Published:** 2021-09-14

**Authors:** Wojciech Zakrzewski, Maciej Dobrzyński, Anna Zawadzka-Knefel, Adam Lubojański, Wojciech Dobrzyński, Mateusz Janecki, Karolina Kurek, Maria Szymonowicz, Rafał Jakub Wiglusz, Zbigniew Rybak

**Affiliations:** 1Department of Experimental Surgery and Biomaterial Research, Wroclaw Medical University, Bujwida 44, 50-345 Wrocław, Poland; adam.lubojanski@student.umed.wroc.pl (A.L.); maria.szymonowicz@umed.wroc.pl (M.S.); zbigniew.rybak@umed.wroc.pl (Z.R.); 2Department of Pediatric Dentistry and Preclinical Dentistry, Wroclaw Medical University, Krakowska 26, 50-425 Wrocław, Poland; 3Department of Conservative Dentistry, Endodontics Wroclaw Medical University, Krakowska 26, 50-425 Wrocław, Poland; anna.zawadzka-knefel@umed.wroc.pl; 4Student Scientific Circle at the Department of Dental Materials, School of Medicine with the Division of Dentistry in Zabrze, Medical University of Silesia in Katowice, Akademicki Sq. 17, 41-902 Bytom, Poland; wojt.dobrzynski@wp.pl; 5Department of Maxillofacial Surgery, Mikulicz Radecki University Hospital, Borowska 213, 50-556 Wrocław, Poland; matjanecki@gmail.com; 6Rajdent, Kozielewskiego 9, 42-200 Częstochowa, Poland; karolinakurek93@gmail.com; 7Institute of Low Temperature and Structure Research, Polish Academy of Sciences, Okólna 2, 50-422 Wrocław, Poland; r.wiglusz@intibs.pl

**Keywords:** nanomaterials, endodontics, dentistry

## Abstract

In recent years, nanomaterials have become increasingly present in medicine, especially in dentistry. Their characteristics are proving to be very useful in clinical cases. Due to the intense research in the field of biomaterials and nanotechnology, the efficacy and possibilities of dental procedures have immensely expanded over the years. The nano size of materials allows them to exhibit properties not present in their larger-in-scale counterparts. The medical procedures in endodontics are time-consuming and mostly require several visits to be able to achieve the proper result. In this field of dentistry, there are still major issues about the removal of the mostly bacterial infection from the dental root canals. It has been confirmed that nanoparticles are much more efficient than traditional materials and appear to have superior properties when it comes to surface chemistry and bonding. Their unique antibacterial properties are also promising features in every medical procedure, especially in endodontics. High versatility of use of nanomaterials makes them a powerful tool in dental clinics, in a plethora of endodontic procedures, including pulp regeneration, drug delivery, root repair, disinfection, obturation and canal filling. This study focuses on summing up the current knowledge about the utility of nanomaterials in endodontics, their characteristics, advantages, disadvantages, and provides a number of reasons why research in this field should be continued.

## 1. Introduction

One of the branches of dentistry that deals with the morphology and physiology of the endodontium is endodontics. It combines such aspects of this field as etiology, pathology, epidemiology, prophylaxis and, above all, treatment of endodontic and periapical diseases. Depending on the complexity of the case, the treatment process may be carried out at one or more visits. Due to the difficulty of maintaining the sterility of the operator’s work area, nanomaterials are increasingly used. Thanks to the expanding variety of nanoparticles, such as bioactive glass, zirconia, chitosan, hydroxyapatite, silver particles, zinc oxide, the properties of materials used in dentistry, such as durability, tissue regeneration and bactericidal properties, can be improved. Biofilm is the cause of many failures in root canal treatment, nanotechnology can provide better bactericidal properties of materials [1]. At the beginning of endodontic treatment, a patient’s medical history should be obtained, followed by a clinical examination. The next step is to take pantomographic and targeted photos, preferably cone beam computer tomography (CBCT), which is the most accurate, but also the most expensive. Only with such medical documentation it is possible to decide on further treatment, including consultations, depending on the case, with specialists such as a dental surgeon, prosthetist or periodontist. Such a complicated procedure is necessary because not every tooth can be cured due to the degree of damage and the complication of the treatment process. The basic procedure in endodontics is the of access to the pulp chamber by means of dental turbines and contra-angles, as well as the chemo-mechanical preparation of the canals with hand or machine tools and rinsing agents [2]. The final stage is the thorough filling of the canals and restoration of the tooth. Infection can occur in many stages, which is why the use of nanotechnology in this field of dentistry is so important [3]. Nanomaterials are characterized by different shapes, such as nanotubes, nanorods and nanowires, offering the possibility of application in various fields of medicine [4]. Their shapes will be described in more detail and visualized in the next chapter of this work. The smaller the size of the particles, the greater their total surface area. Dentistry is a rapidly developing field of medicine. In the case of endodontics, nanotechnology ensures wide application in sealers, obturating, composites and root repair materials or disinfectants. Nanotechnology is also applicable to the coating of dental instruments, nano sized drug related therapy and pulp regeneration. Especially in endodontics, it is important to use new technologies in tools and materials to improve the procedures and increase the percentage of correctly performed endodontic treatments. Drug delivery systems are a promising field of nanotechnology, which has recently attracted more and more interest in dentistry, both in the case of organic and metallic nanoparticles. Therefore, further research into nanotechnology is important for finding the most suitable nanoparticles for dental applications. Thanks to its versatility, nanotechnology can provide dentists with greater comfort at work [5]. This work collects the latest information related to the use of nanotechnology in endodontics, shows how great its potential is and how important further research in this field is, to increase the effectiveness of treatment in dentistry.

## 2. Classification of Nanomaterials, Materials Modification

Nanomaterials are natural or manufactured materials. They have regular structures containing molecules in such an unbound state, in which at least half of the molecules have the molecular level that does not exceed the size of 100 nm [6]. Nanoparticles with their improved and exclusive properties, including extremely small size, elevated chemical reactivity, as well as high surface to mass ratio. With such features, they have led to research into new perspectives for effective dental treatment [7]. The authors suggest that the different features of these materials are connected with the correlation between the increased surface ratio and increased number of atoms at the surface of such material. These specific characteristics can be used to prepare very specific devices and biomaterials for interaction at the both cellular and subcellular levels of the human body in order to reach maximal therapeutic effectiveness [8].

In addition, large surface area of nanomaterials leads to an increased potential of interacting with biomolecules, eventually forming a corona [9]. It is a nonspecific interaction causing changes in NPs biological and physicochemical properties [10].

Nano-particles are used in endodontics mainly to eradicate microorganisms in the root during root-canal and restorative treatment procedures. The research regarding improvement of nanomaterials is focused on improving their antimicrobial efficacy [7,11,12], the mechanical characteristics of the dentin [13], eliminating biofilm [14] and dentin hypersensitivity [15] and regenerating tissues [16].

Generally, nanoparticles can be divided into naturally occurring and artificial due to their composition. The subgroup of naturally occurring nanoparticles divides into inorganic or organic. As for the shape, nanoparticles can be divided into spherical, tubular, rod-shaped and plate-shaped particles. In addition, functionalized nanoparticles can also be distinguished. They have an inner part-core, that is built of one material with different molecules on its outer surface or enclosed in it. Depending on application, particles in nanomaterials can be modified by using, among many things, drugs or peptides [17]. The mechanism of functionalization is focused on the functioning of linker molecules in which each linker molecule has at both its ends a reactive group that binds different molecules—such as biocompatible materials (dextran), antibodies, fluorophores and others—to the core of the nanoparticle. On the other hand, the core nanoparticle can be used as a surface for assembling particles from inorganic or organic materials [1].

Another classification method for nanomaterials is their division by dimensions, as illustrated in Figure 1; materials can be produced in a nanoscale in either zero (e.g., fullerenes), one (thin surface coatings), two (e.g., graphene), or three dimensions (composite nanomaterials).

In this study, the authors would like to describe and discuss two main types of nanomaterials, named bioactive and bioinert. Examples of the former are bioglass materials or hydroxyapatite. Tricalcium phosphate is also an important part of the group. The latter are represented by zirconia, alumina, and titanium. New trends and future prospects in the field of nanomaterials in dentistry will be discussed as well.

### 2.1. Quantum Influence on Nanomaterials

#### 2.1.1. Quantum Confinement Effects

When it comes to nanomaterials, it is important to underline that quantum effects are not dominant in macro- and micro-sized materials, even though they are present in the nano scale. Quantum effects may affect the electrical, magnetic and optical behavior of materials.

Roduner et al. [18] point out why nanomaterials are different from their larger-in-scale counterparts. One of the many differences is the catalysis processes on their surfaces. It changes the chemical, physical, as well as electronic properties. Quantum confinement effects describe electrons in terms of electron energy band gaps, conduction bands, valence bands, potential wells and energy levels [19]. It is present when a particle is too small to be comparable with electron wavelength. This is a general condition that is strictly limited by specific material properties, especially Bohr radius [20]. The density of states (DOS) can be explained as a model of ‘particle in a box’, where the size of the particle is directly proportional to the size of the box [21]. Roduner et al. [18] also point out the role of pseudo-atoms in clusters of metal particles influencing the characteristics of the whole material. Generally, the magnetic effect is caused by movements of particles with both electric charge and mass. Spinning movement results in forming a magnetic dipole called magneton. It is then clustered in groups within a ferromagnetic material [22,23]. Originally nonmagnetic elements that become magnetic can be an interesting feature of nano-sized particles. Rodunet et al. [18] suggest that nano-size may influence the magnetization of particles which may become clinically useful in the future. It is also possible that the scale of nanomaterials may result in different properties of nanoparticles when compared to their larger counterparts, for example copper may be transparent [24]. This example alone shows that if research into nanomaterials continues, the potential for their use may turn out to be much greater than initially thought. This review only mentions the quantum influence on nanomaterials, but there are a plethora of scientific works that can expand the knowledge needed for a better understanding of quantum effects [25,26,27,28].

#### 2.1.2. Surface Effects

Nano-sized materials have novel characteristics that are absent in the behavior of molecules forming a bulk [29,30,31]. When the number of molecules in the nano-scale is reduced, the materials usually reach a point at which the whole substance starts to behave in a way which is more characteristic for molecules than for a bulk matter. To understand the thermophysical properties of nanomaterials better, theoretical research is required [32,33,34,35,36]. The general rule is that mechanical and thermodynamic parameters are reduced with decreasing size of particles. Among many features, it applies to Young’s modulus [37], mass density [38], cohesive energy [39,40] and melting point [41,42,43]. Macroscopic thermodynamics does not apply to nanomaterials because of the differences in the binding energy that affects them. As L. D. Gelb et al. [44] confirmed that phase transition is less visible in nano-phase than in macro-phase. Instead of using the concept of ‘phase’, it can be more correct to speak of ‘different structural isomers’ that coexist over a range of temperatures [45].

Xiaohua et al. [29] showed that theoretical analysis or experimental data can be used to predict relations between thermodynamic and mechanical properties.

### 2.2. Chitosan Nanoparticles

Chitosan is an organic chemical compound from the group of polysaccharides. This non-toxic biodegradable substance is obtained from the deacetylation of chitin in alkaline media. Sources of this homopolysaccharide are the exoskeletons of arthropods, mollusks and insects. Chitosan is a polysaccharide containing deacetylated units and acylated units. These unit are, respectively: D-gucosamine and N-acetyl-D-glucosamine. Due to its availability in many forms such as hydrogels, capsules or scaffoldings, it of interest as a material for biomedicine [46]. In general medicine, it is useful as a bandage mimics the native extracellular matrix, ensuring the appropriate microenvironment of the wound, thus accelerating its healing. In endodontics, chitosan can be used mainly due to its antibacterial properties [47], especially against *E.faecalis* strain [48,49].

Regarding its unique chemical structure, it has many applications (alike only or as an alternative combine with other natural polymers), such as medical use, cosmetic use, also bio-printing, agricultural and horticultural use [50]. In biomedical engineering chitosan nanoparticles are being used in view of physicochemical properties such as, particularity, biocompatibility and sensitivity [51].

The solubility of chitosan in the acidic environment is responsible for its antibacterial activity. As a result, negative charges interact with the cations formed during the protonation of the C-2 amino groups. Amino groups disorganize membrane structures when interacting with the negative charges of cell membranes and as a result provoke destruction of microbes [52]. Factors such as degree of substitution, molecular weight and degree of deacetylation have an influence on the antimicrobial properties of chitosan and its derivatives [53]. It is worth noting that chitosan combined with 2% chlorhexidine gel showed very good destructive effects against *E. faecalis* in root canals [54]. This mixture also works well as a root canal sealer [55].

### 2.3. Hydroxyapatite (HAp)

The naturally occurring mineral form of calcium apatite is hydroxyapatite (HAp). It is mainly obtained from mineral tissue [56]. Hydroxyapatite is highly biocompatible due to the fact that HAp is one of the main constituents of dentin. In addition, it can also quickly osseointegrate with the supporting connective tissue (bone tissue). HAp is used in a variety of forms, equally as composites, coatings [57] and powders [58] for dental fillings in view of these advantages.

The high bioactivity of nanohydroxyapatite results from its similarity to bone apatite as well as its strong ion exchange affinity. In addition, its biological properties are determined by the size and morphology of molecules, the type of ionic impurities in the crystal lattice and the Ca/P molar ratio [59,60], resulting in its hexagonal system crystallization [61]. In addition, its biological properties are determined by the size and morphology of the molecules (Figure 2).

After accidental exposure of the pulp, nanohydroxyapatite may be used as a substitute for calcium hydroxide and MTA in the direct capping method (biological treatment). This is because of its bioactive properties (tissue stimulation through the reparative formation of dentin), as well as antimicrobial and mechanical properties (HAp withstands effort generated during the placement of restorative materials) [62]. It can be effective in procedures such as augmentation of the periodontal bone defects or alveolar ridge [63].

Benita Wiatrak et al. [64] showed that nanohydroxyapatite molecules can also induce nerve regeneration. There is an interaction between the time of nerve inflammation and an increase in the activity of mitochondria in neurons affected by hydroxyapatite nanoparticles. In dentistry, nano-hydroxyapatite also plays a significant role due to its remineralizing properties. Remineralization occurs through the precipitation of calcium and phosphate ions, which are provided by hydroxyapatite. This phenomenon is observed with the partial demineralization of the collagen matrix [65,66,67]. Rana S. Al-Hamdan et al. [65], as well as Allaker et al. [68], showed that nHAp particles are added to the bond to improve the biomechanical properties of the adhesive resin. Moreover, this mixture strengthens the structure and durability of the tooth. Nanophase hydroxyapatite has a smaller contact angle and a greater number of surface atoms compared to nanohydroxyapatite in the microscale. Additionally, it also has such unique properties in contrast to the conventional microsized hydroxyapatite, including greater surface area and altered electronic structure. The polymer matrix has better mechanical properties by adding hydroxyapatite nanoparticles to it. Adding nHAp to chitosan scaffolds mRNA upregulation for Runx2, ALP Smad1, BMP-2/4, collagen I and integrin subunits along with myosins and also increases bone marrow stem cell proliferation relative to the addition of Hap microns proved in their research by Liu et al. [69]. Other studies of this type have shown that nanohydroxyapatite is used to improve bioactive properties and to strengthen polymer networks [70]. Furthermore the addition of nanohydroxyapatite enhances spread and cell adhesion. Bone reconstruction and periodontal regeneration also take place by injecting the mentioned bioactive materials constituting the matrix of the entire process. However, despite its various advantages, hydroxyapatite has poor mechanical properties and therefore cannot be used for load-bearing applications [71]. In order to enhance nHAp characteristics, it can be doped with elements such as metal ions. It has been confirmed that, as a result of this action, several characteristics can be influenced, including the biodegradation rate reduction [72]. Summing up, nHAp-reinforced nanocomposites improve the mechanical rigidity and bioactivity of implants. Additionally, they can be used to rebuild the tooth due to its unique properties.

### 2.4. Bioactive Glass

In 1960, Hench et al. developed bioactive glass which consisted of strictly defined proportions: sodium oxide, calcium oxide, phosphorus pentoxide and silicon dioxide. This material is widely used in tooth repair due to its ability to bind to bones [73]. Additionally, bioactive glass has excellent regenerative and antimicrobial properties. Its structure allows new bone tissue to be regrown directly onto it, as is visualized in Figure 3. The process of bone tissue regrowth is possible due to the similar chemical composition of bioactive glass, human bone and dentin.

Bioactive glass is successfully used in dentin remineralization [74,75]. This is possible through the precipitation of minerals through solubility upon contact of bioactive glass with human plasma or saline. As a consequence, hydroxyl carbonate apatite (HCA) crystallizes at the glass/tissue interface. Bioactive glass is more effective in sealing dentin. This process is effective due to the hydrophilic properties of the materials and the consequent expansion of moisture towards the canal wall [76,77].

Bioactive glass also has antibacterial properties. This is possible when several factors work together [78]. These include high pH, Ca/P precipitation and osmotic effects.

The increase in pH occurs when the bioglass is dissolved in water and thus releases ions.

In turn, precipitated Ca/P ions initiate mineralization on the surface of the bacteria, and an increase in the osmotic pressure above 1% inhibits numerous bacteria.

### 2.5. Zirconia (Zr)

Zirconia (Zr) is a stabilized regular modification of zirconium oxide. This material is characterized by: high wear resistance, good optical properties and low reactivity. As a result of its properties, it is used in implantology, as well as in dental restorations [79,80].

Endodontic treatment causes the reduction of the tooth’s mineral tissue (its loss) and consequently, weakening of the tooth. The reduced amount of tooth tissue makes its reconstruction difficult. Core buildups are used to retain core materials that are predictably delivered with state-of-the-art resin composites [81,82]. Therefore, the core material (its mechanical and physical properties) is important in the endodontic restoration of the tooth. To strengthen the structure of the weakened tooth, nanoparticle-sized fillers for composites are used. The great biocompatibility and mechanical properties of zirconium nanoparticles make it enriched with composite resins.

Based on laboratory tests, it can be observed that reconstructing composites containing Zr nanoparticles showed better compressive strength compared to other composites and compared to micro- and macromolecules based on silica and barium [83].

### 2.6. Nanosilver

The antimicrobial, anti-inflammatory, thermal and optical properties of silver made it popular in medicine [84]. In dentistry, silver is mainly used for its antimicrobial properties. The rate of silver ions release determines its unique antibacterial properties. Silver is considered fairly inert even though it exhibits metallic properties and it becomes highly reactive due to the fact that it is ionized with moisture. The antibacterial properties of silver are related to the fact that it binds to tissue proteins of the cell wall, thus changing its structure and, consequently, destroying the bacterial cell [85]. The antimicrobial and anti-adhesive properties of silver nanoparticles coated with polyvinyl alcohol and farnesol were assessed against *E. faecalis*, *C. albicans* and *Pseudomonas aeruginosa (P. aeruginosa)* were assessed by Chávez-Andrade et al. [86]. These studies confirm the effectiveness of the use of silver nanoparticles as an adjunct to endodontic treatment in root canal disinfection or for the inhibition of biofilm formation [86]. The research by Juan M Martinez-Andrade et al. [87] revealed that nanosilver can be added to EDTA 17% for rinsing root canals as a new modification. The in vitro studies described above prove that EDTA-AgNPs can be used to effectively remove the smear layer while exhibiting antimicrobial activity during root canal treatment.

### 2.7. Zinc Oxide (ZnO)

Resin-based root canal sealants can be enriched with nanoparticles of, i.e., ZnO. Enrichment of sealants with zinc oxide nanoparticles improves antimicrobial properties through better diffusion of root canal sealants. Research has shown that endodontic microorganisms are closely related to gutta-percha (a common material for filling root canals) [88,89]. Often the cause of infection in the root canal is the adhesion of bacteria and the formation of biofilms on the gutta-percha. In such cases, increased effectiveness of root canal disinfection could be possible due to the long-term antimicrobial properties of root canal sealants. In most cases of the sealants used, a decrease in the antimicrobial properties is observed immediately after binding. However, those commonly used retain its antimicrobial properties for a maximum of 1 week after binding [90,91]. A high percentage of *E. faecalis* bacteria reduced the adhesion to the dentine by treating the dentine of the root with ZnO nanoparticles, ZnO/CS mixture, ZnO of the CS layer or NP CS [92]. Magnesium oxide and calcium oxide suspensions have a bactericidal effect on both gram: positive and negative bacteria. On the other hand, zinc oxide suspension has a stronger antibacterial effect against Gram-positive than Gram-negative bacteria, and also has a bacteriostatic effect [93].

Zinc oxide is mainly used for its antimicrobial properties. The process takes place through the electrostatic interaction of nanoparticles with the bacterial cell membrane (nanoparticles are charged positively, and the membrane is negatively charged). As a consequence of the accumulation of nanoparticles, the permeability through the membrane is inhibited, which is a further cause of bacterial cell death [94].

### 2.8. Exosomes

Bioactive vesicles secreted by various cells into the extracellular environment are called exosomes [95]. Their size ranges from 30 to 100 nm spherical membrane nanovesicles. Apart from apoptotic bodies and ectosomes, they belong to a common group of membrane microbubbles (extracellular vesicles, EV). The criteria of division and separation of this group is based on their size, physicochemical properties, as well as the mechanism and sources of formation.

In general, exosomes are involved in intercellular communication and are produced by mesenchymal stem cells. They are characterized by greater biocompatibility and stability compared to mesenchymal stem cells (MSC). In addition, they are characterized by lower immunogenicity, and at the same time they are easier to manufacture, transport in clinics and easier to store [96]. Importantly, these are vesicles that occur in a variety of body fluids and can be secreted by most types of cells [97].

The delivery of bioactive molecules by exosomes to various target cells makes them participate in various processes such as tissue regeneration and angiogenesis. In addition, primary vesicles may also be involved in the immune response, inflammation and apoptosis [98].

The properties of exosomes may be used in the future to regenerate the dental pulp.

Various MSCs participate in the process of pulp formation, such as dental pulp stem cells (DPSC), human exfoliated deciduous tooth stem cells (SHED), apical papilla stem cells (SCAP), and dental alveolar progenitor cells (DFPC) [99]. Zhou et al. [100] proved that microballoons from dental pulp stem cells (DPSCs) can be used in regenerative endodontic therapy due to their pro-angiogenic effect.

As a result of exosomes activity that emerges in these cells, pulp regeneration is possible. Some of them also enable bone and periodontal regeneration, such as cells isolated from human milk teeth (SHED), which have been shown to have the ability to osteogenic regeneration.

In a study by Maria Giovanna Gandolfi et al. [101], Polylactic acid (PLA), dicalcium phosphate dihydrate (DCPD) and calcium silicates (CaSi) were doped with exosome vesicles (EV) to test their effectiveness in bone regeneration. The tests showed that the exosomes present in the experimental materials improve the expression of genes: osteocalcin, osteopontin, osteonectin and type I collagen (the main markers of osteogenesis). As a consequence, they prove to provide better bone regeneration which makes exosomes to have a great potential for future dental regeneration procedures.

### 2.9. Graphene

Graphene is a two-dimensional single layer of sp2 hybridized carbon atoms [102]. It has a hexagonal configuration and its pattern resembles a honeycomb.

Different surface properties, size and number of layers make it possible to divide the graphene into several groups [103]. The common group include: reduced graphene oxide (eGO), graphene oxide (GO; monolayer up to several layers) and graphene nanocards (GNS). In addition, this group includes multi-layer gafen (FLG), graphene nanoplates (GNP) and ultra-thin graphite [104].

Graphene has many properties that can be used in dentistry, including antibacterial activity, potential use of graphene in tissue engineering, dental implants, in the field of endodontics, periodontics and conservative dentistry.

Graphene nanoplate exhibits antimicrobial properties due to the sharp edges of GNP flakes piercing the soft cell wall of the bacteria, resulting in their trapping and shrinkage [105]. These results suggest that *Streptococcus mutans* (*S. mutans*), which is responsible for human caries, can be combated in this way.

Graphene used as a coating on a titanium implant has the potential to improve the properties of osseointegration, accelerate tissue healing, and inhibit microbial growth [106]. It was tested that graphene in combination with silver nanoparticles shows strong antibacterial properties with a lower cytotoxic effect on soft tissues and bones compared to 3% sodium hypochlorite [107]. This makes graphene a promising material for infected canals irrigant in the future.

In endodontics, apart from root canal rinsing, it is also possible to add it to bioactive cements such as Biodentine. This action is to shorten the setting time of the cement by reducing induction and accelerating the hydration process [108].

In conclusion, it is worth developing research on graphene in the future, due to its unique properties along with independent functioning, as well as in combination with other biomaterials, which may expand its clinical application in dentistry due to its biocompatibility and antibacterial properties.

### 2.10. Nanopolymers

Quaternary polyethylene ammonium (QPEI) is a nanopolymer that exhibits insolubility, biocompatibility and chemical stability in contrast to other materials used in chemo-mechanical root canal disinfection such as calcium hydroxide materials [109]. The studies conducted by Abramovitz et al. show, that nanopolymers of quaternary ammonium polyethyleneimine (QPEI) have a long-lasting antibacterial activity against both Gram-negative and Gram-positive bacteria [110].

When added to the epoxy resin sealant, this material reduces the viability of *E. faecalis* in dentinal tubules. As a consequence, endodontic treatment is more effective and thus the likelihood of complications caused by bacteria affected by QPEI is reduced.

Manuel Toledano et al. [111] assessed the effectiveness of polymer nanoparticles doped with calcium (Ca-NP.), Doxyckline (D-NP.) and zinc (Zn-NP.), among others. Experiments have shown that the highest sealing efficiency was demonstrated by Zn-NPs with the highest values of Young’s modulus and dentin mineralization.

Remineralization of dentin in endodontically treated teeth has proven to be facilitated due to the nanoparticles provoking easier precipitation and production of amorphous calcium phosphate precursors. Besides, nanoparticles are able to bind with collagen [112].

Based on these results, it is concluded that polymers doped with zinc nanoparticles mechanically strengthen the root dentine. Additionally, they close dentinal tubules by precipitating crystalline calcium phosphate. Therefore, it is encouraged to use this material prior to root canal filling in endodontic treatment. It should be emphasized that filling microcracks and cavities with the use of endodontic cements and sealants is obligatory to ensure the effectiveness of treatment in root canal therapy [113].

## 3. Clinical Applications

### 3.1. Sealers

Despite the advanced studies on improving the properties of gutta-percha, it has not been possible to eliminate the need to use a sealer for three-dimensional filling of the canal system. Current research shows that sealer is more important than the study material [114].

The crucial functions of an endodontic sealer are to make the seal impervious. This can be provided by filling in the minor distortions and irregularities that may occur between the root canal wall and the stem filling material. In addition, if the microorganisms remain in the lateral canals or tubules, the sealant then fulfills the microbiological control. Studies have shown that bacteria that occur in root canals survived at 40–60%, despite the use of different concentrations of NaOCL in the chemo-mechanical method of root canal cleaning [115]. Most sealants show mild antimicrobial properties. This is a consequence of the release of eugenol, paraformaldehyde or zinc oxide. However, these properties gradually diminish as the sealant sets [116]. The quality of the root canal filling is greatly influenced by the thickness of the endodontic sealant layer. Only a thin layer should be evenly applied to the canal walls because it shrinks during setting and leaves unwanted voids. Moreover, these sealers gradually dissolve [117].

Silver (Ag), calcium oxide (CaO), copper oxide (CuO), zinc oxide (ZnO), chitosan (CS), magnesium oxide (MgO) and QAPEI nanoparticles have been investigated as potential antimicrobial agents and to improve physicochemical and biological properties.

Epoxy resins are among the materials that are the most popular with clinicians because of their high pressure resistance, good marginal adhesion and minimal solubility in tissue fluids. They also exhibit antimicrobial activity and can form chemical bonding with dentinal collagen [118]. Moreover, they are characterized by a long working time and good availability. Significant disadvantages of these materials are the difficulty of removing them from the root canal system and some degree of cytotoxicity, which may vary [119].

Barros et al. [120] proved by conducting the test that, by adding 1% or 2% nanoparticles of QPEI, it influences the antimicrobial effect of sealants (in particular, with oral cavity bacteria). Adding a nanoparticle to the sealant does not significantly affect compressive strength, dimensional change, solubility, flowability, apparent porosity or change of setting time. QPEI has a positive charge and is hydrophobic—after its incorporation, sealers become hydrophilic. Therefore, it is possible to use it as an antibacterial biomaterial. and QPEI nanoparticles enhanced antibacterial effectiveness against *E. faecalis* strains. After seven days, no significant changes in mechanical and physicochemical properties were noticed.

Beyth et al. [121] observed that antibacterial effects against *S. mutans* and *E. faecalis* are obtained mainly by adsorption and penetration through the bacterial cell wall. In the next stage, it interacts with proteins and the fat layer in the cell membrane. Subsequently, the exchange of essential ions is blocked, the cell membrane is destabilized and, consequently, cell death occurs [122]. Physicochemical and mechanical tests carried out by Barros et al. showed that it is possible to increase the penetration capacity of the sealant into the root canals by adding QUPEI nanoparticles to it. The type of sealant and the type of cell determine the proliferation and differentiation of bone cells without increasing the cytotoxicity of the sealant. Scientists have demonstrated this by adding 2% of QPEI molecules to AH Plus and PCS.

A material that prevents later reinfection and is useful in the control of endodontic infections during root canal filling has been demonstrated by Gong et al. [123]. AH Plus based on epoxy resin has been enriched by scientists with quaternary ammonium epoxy silicate (QAES) having a rough surface, spherical shape and a diameter of about 120 nm. Copolymerization of the polymer network of epoxy resin materials with QAES is possible through epoxy groups of these molecules. The authors have found it effective in in vivo root canal disinfection after obturation; hence, the possible benefit for successful endodontic treatment have been shown, especially regarding the reduction of the *E. faecalis* planktonic bacteria and monospecies biofilms. Activity against the colony of *E. faecalis* was also investigated on the strength of various nanoparticle sealers such as nanochitosan, Ag, ZnO, nano-calcium hydroxide [124].

#### Chitosan as a Sealer

Chitosan is used in endodontic treatment, not only because of its biocompatibility but also due to its outstanding disinfecting and chelating effects. This organic compound is mainly used because of its wide range of antimicrobial activity. Chitosan affects such microorganisms as yeasts, filamentous fungi, Gram-negative bacteria and Gram-positive bacteria [125]. Nanoparticles of chitosan are mixed with sealers based on epoxy resins or calcium hydroxide to modify their effectiveness. The antimicrobial process is based on electrostatic interaction (connection with negative electric charges of the bacterial cell membrane). This interaction causes the cell membrane to increase its permeability. As a consequence, leakage of intracellular components and cell death occurs [126].

In the studies of Nair et al. [127] to analyze the anti-biofilm effectiveness of sealants containing nanoparticles, confocal laser scanning microscopy (CLSM) and the colorimetric crystal violet test were used. A sealant based on chitosan nanoparticles and calcium hydroxide as well as a sealant with the addition of zinc oxide nanoparticles has been tested by scientists in terms of anti-biofilm effectiveness. Maintaining a high pH level and the release of hydroxyl ions ensure their high antimicrobial activity; however, their effect on *E. faecalis* is limited. *E. faecalis* express a proton pump in their plasma membrane for energy metabolism and survive the high alkaline pH.

In an experiment by Nair et al. [128], an attempt was made to modify a calcium hydroxide sealer (Apexit Plus) to improve its antibacterial properties against two strains of *E. faecalis*: OG1RF and ATCC 29212. The significantly increased antibiofilm effect against the ATCC 29212 *E. faecalis* strain and the questionable effect against the OG1RF strain are shown by the inclusion of nanoparticles (CS and ZnO) in calcium hydroxide sealants according to research conducted by scientists.

A. Del Carpio-Perochena et al. also assessed the effectiveness of chitosan nanoparticles incorporated into calcium hydroxide paste. In tests, they showed that a significant reduction of colony forming units of E. faecalis (CFU) is possible with the calcium hydroxide paste integrated with CNP after 7 and 14 days of application. Additionally, the inhibition of bacterial recolonization after endodontic treatment was observed in this test. To sum up, the application of CNPs into Ca(OH)_2_ pastes may be beneficial and makes them an attractive alternative to modern canal filling agents [129].

Nanoparticles of chitosan were also used to enrich bioceramic root canal sealers. The effective antifungal and antibacterial activity of these materials is possible due to their physical and biological properties. In addition, they can contain calcium phosphate. This additional material brings the chemical composition and crystal structures close to the apatite materials of teeth and bones, thus improving the adhesion of the sealant to the dentin. In addition, chitosan nanoparticles improve the binding properties of bioceramics [130]. However, they also have disadvantages such as difficulty in repeated endodontic treatment resulting from their difficult removal from the root canal, as well as in the preparation of the space after the procedure [131].

### 3.2. Obturating Materials

Removal of infected tissues from root canal systems, their disinfection and tight filling with proper protection of the tooth crown are the standards of the modern approach to endodontic treatment. After removing the content of the tooth cavity and shaping the canal properly, the root canal should be hermetically filled in three dimensions to prevent being recolonized by microorganisms. Moreover, it provides the isolation of any pathogens from the saliva that are responsible for percolation from the coronal region, the gingival sulcus or periodontal pockets. Only hermetic filling from the coronal orifice to the physiological foramen protects tissues from disease processes, and in the case of existing inflammations, it determines their healing. Obturation is a critical moment and is often the reason why endodontic treatment fails. Several techniques and materials have been developed to provide the cohesive and adhesive interaction of filling materials with the walls of the root canal system. The properties required of perfect filling materials used in endodontic treatment are biocompatiblility with the tissue, stimulation, or at least support of the regeneration of damaged cells, easy adaptation to and removal from the canals, antimicrobial activity, adherence to the canal walls and dimensional stability over time, water insolubility, good handling and flow characteristics, radiopacity, imperviousness and non-porosity, both local and systemic nontoxicity, lack of tooth discoloration, reinforcement, support and strengthening of the root structure as well as affordability and a long shelf-life [132].

Currently, despite the advances in materials engineering and molecular biology, no material has been developed that would meet all of these requirements. A wide range of them have been presented over the past decade for root canal filling, from gold and silver cones, gutta-percha, Resilon, calcium phosphate, MTA and various types of sealers.

The basic concept of three-dimensional root canal filling involves the use of two materials at the same time—the main one acting as a core with solid or semi-solid consistency, and the other with a semi-solid consistency acting as a sealer, filling the space between the dental wall and the obturating core interface and flowing into multiple canals, fins, deltas, and lateral canals. The most commonly used material for the main filling is gutta-percha. It is used both in cold filling methods including a single gutta-percha cone and lateral compaction, or warm filling methods such as warm vertical or lateral compaction, thermoplastic injection technique, root canal filling with a gutta-percha coated carrier, continuous wave compaction technique. Research shows that gutta-percha alone, even thermoplasticized, does not provide adequate three-dimensional filling and should be used with sealers [133]. This material, despite many advantages such as biocompatibility, inertness or malleability, cannot provide the desired hermetic seal and inhibits coronal microleakage along with the penetration of microorganisms and their proliferation in the volume of voids and irregularities in root canal system.

Gutta-percha is a thermoplastic polymer of natural origin. Chemically it is trans-1,4-polyisoprene and can exist in different crystal forms—alpha and beta—as well as in the isomeric-amorphous gamma phase. In dentistry, the most commonly used form is the beta configuration in the form of a gutta-percha cone, which is stable and flexible at room temperature and is less adhesive and flowable when heated [134]. Alpha is used in thermoplastic methods of filling root canals, e.g., Thermafil^®^ (Dentsply Sirona, Charlotte, NC, USA), and is adhesive and highly flowable when heated and brittle at room temperature. For endodontic purposes, gutta-percha cones are used in various sizes and the composition may differ slightly depending on the manufacturer. The clinical features of gutta-percha are its easy removal from the root canal, good X-ray contrast, stability in volume, not being affected by humidity and not causing tooth discoloration. Research indicates that it shows some antimicrobial properties, probably related to zinc oxide in its composition [135]. Numerous in vivo and in vitro studies show some cytotoxicity regarding the release of zinc ions may exert toxic effect to the surrounding tissues. It is unlikely that gutta-percha alone will induce symptoms in allergic patients [136].

Regarding the lack of true adhesion of GP and consequently microleakage, root canal reinfection, and not enough completed mechanical properties, attempts were made to use additives to improve these capacities [137]. Researchers have investigated the medical activity of gutta-percha cones containing different substances including calcium hydroxide, bioceramic, resin, iodoform, zinc oxide, nonthermal plasma-argon, and oxygen plasma chlorhexidine, and cetylpyridinium chloride alone or used in combination [134,138].

Scientists have also explored nanodiamonds (NDs) and nanosilver particles as an alternative modification to obtain optimum sealing and therapeutic effects [139]. Gutta-percha coated with nanosilver particles (AgNPs) exhibited antiseptic properties. After modification, this combination has shown antibacterial activity against *E.faecalis*, *Staphylococcus aureus (S. aureus)*, *Candida albicans (C. albicans)* and *Escherichia coli (E. coli)*. In in vitro studies on mouse fibroblasts, Shantiaee et al. [140] indicated that this material is biocompatible and has cytotoxicity like standard gutta-percha. In their investigation, it achieved the lowest level of cytotoxicity among the tested materials after one week. In vitro studies demonstrated by Yazdan et al. [141] also suggested that the antibacterial effect of nanosilver coated gutta-percha may be more effective in endodontic treatment compared to standard gutta-percha. They compared for 60 days standard gutta-percha and nanosilver coated gutta-percha by dye and bacterial leakage methods indeed there was no significant difference between the two fillers either in bacterial or in dye leakage results. Lee et al. [137], in order to reduce the probability of root canal reinfection, improved mechanical properties by proposing a nanodiamond-gutta percha composite (NDGP) functionalized with amoxicillin (ND-AMC). The combination of ND with an antibiotic may affect the elimination of bacteria remained after the endodontic treatment. In addition, the research showed broad bactericidal properties against pathogens moving from the side or additional canals. It has been indicated that NDGP increases the effectiveness of treatment and can be used together with conventional endodontic therapy procedures [137].

Carbon nanoparticles of 4 μ–6 nm in diameter were used as a platform for drug delivery. Because of their favorable chemico-physical parameters such as versatile faceted surface, lack of cytotoxicity, and their ability to improve the mechanical properties of other materials [142]. In their studies Lee et al. used Amoxicillin as a broad-spectrum antibiotic which is the first line of antibiotics chosen as adjunctive systemic antibiotic treatment during endodontic therapy. Researchers have shown that carbon nanoparticles exhibit physicochemical properties enabling electrostatic adsorption and/or covalent conjugation of numerous compounds [142], they also demonstrated antimicrobial activity.

Bioceramic materials are an example of materials based on tricalcium phosphate, mineral trioxide aggregate (MTA) and tricalcium silicate. In research by Del Carpio-Perochen et al., 1 g/150 mg chitosan nanoparticles were added into epoxy resin and calcium silicate sealers, including ThermaSeal^®^ (Dentsply Sirona, Charlotte, NC, USA) and MTA Fillapex^®^ (Angelus, Londrina, Brazil). They showed significant improvement in the antibacterial properties of Thermseal and MTA Fillapex against *E. faecalis* after the incorporation of CNps. The authors also conducted studies using the root canal infection model, investigated the effect of carboxymethyl chitosan (CMCS) or CMCS + rose bengal filled with gutta-percha and CNPs incorporated sealers on the formation of *E. faecalis* biofilm. After four weeks, material with CNPs was characterized by lower bacterial colonization under confocal laser scanning microscopy compared to the control sample, where high viable biofilm total volume at the sealer-dentin interface was observed [47].

For an endodontic treatment to be predictable and effective, it is also necessary to eliminate *C. albicans* as the pathogen responsible for the causes of endodontic failure. S Pattanaik et al. investigated the performance of epoxy resin sealant—AH Plus^®^ (Dentsply Sirona, Charlotte, NC, USA), calcium hydroxide sealant—Apexit Plus^®^ (Ivoclar Vivadent, Schaan, Liechtenstein) and sealant MTA Fillapex^®^ (Angelus, Londrina, Brazil) with and without the addition of 2% wt/vol chitosan nanoparticles on the optional *C. albicans* anaerobic microorganism. The study indicated that these nanoparticles had superior antifungal properties than the sealer itself. It was proved that AH Plus sealer integrated with chitosan manifested better antimycotic properties when compared to Apexit Plus and MTA Fillapex [143]. In conclusion, a combination of chitosan nanoparticles and mixing chitosan with an endodontic sealant offers additional benefits.

Modifications to nanoparticles of chitosan, for instance by functionalization with rose-bengal, are also possible. Thanks to the use of nanomolecules incorporated with various ligands or peptide, it has become possible to undertake new strategies to counter bacterial resistance [144].

In a study by A. Shrestha et al., CSRBnp’s (Bioactive polymeric chitosan nanoparticles functionalized with rose-bengal) by adhering to the bacterial cell surface, permeabilizing the membrane and lysing the cells subsequent to photodynamic treatment, enabled single-session endodontic treatment. CSRBnp’s mechanism of action ensured the elimination of eliminate bacterial-biofilms and stabilize dentin-matrix [13]. As a result, the viability of *E.faecalis* is reduced and the structure of the biofilm is disrupted.

Metal oxide nanomolecules were also studied as potential agents for root canal obturation. To enhance the physicochemical and antimicrobial abilities, zinc oxide NPs were applied to improve materials used during Root Canal Treatment Procedures. In the experiment, Javid M. proved that sealer associated with different degrees of zinc oxide nanoparticles are appropriate for being used in endodontic treatment to avoid bacterial leakage and tend to achieve better results in comparison with AH 26TM and micro-zinc oxide eugenol sealer.

Zinc oxide eugenol-based types of cement are generally applied in endodontic materials. Most of them are in the form of a powder (zinc oxide), which is obtained from eugenol. To reduce porosity and increase the stability of the filling and contrasting compounds, various additions have been applied.

The advantage of zinc-oxide-eugenol sealer is its antimicrobial activity, the ease of filling the space between the gutta-percha cone and the root canal walls, whereas the disadvantages of this group of sealer are its long setting time, shrinkage on setting, dissolution in tissue fluids, and the possibility of staining the tooth structure. Eugenol, which is found to leak from zinc oxide eugenol sealers, induces adverse effects and decreases the transmission in the neurons even after being cured [145].

In the case of using zinc oxide nanoparticles, changes in physicochemical properties were noticed, including setting time improvement, dimensional stability, flowability, solubility, and radiopacity of Grossman sealer in compliance with ANSI/ADA requirements [146]; reduction in cytotoxicity was also observed and it was demonstrated that NZOE exhibited lower cytotoxicity to other commercially available sealers, such as Pulpdent and AH-26 on HGF, and may be a potential material for use in endodontic treatment [147].

Omidi S. et al. [148] examined the tissue reaction to the novel nano zinc-oxide eugenol (NZOE) sealer. Researchers compared the Pulp Canal Sealer (ZOE based) and the AH-26 (epoxy resin sealer) on a rat model. Polyethylene tubes containing the test materials were transferred into the backs of Wistar rats. After the animals were sacrificed, the implants were removed from the surrounding tissues and compared for the presence of inflammatory cells. The zinc-oxide-eugenol sealer had no significant differences in comparison with a commercial ZOE sealer (Pulp Canal Sealer) and an epoxy resin sealer (AH-26). All the implanted materials showed good tissue tolerance and had acceptable biocompatibility.

Studies by H.H. Cheng et al. on experimental carbon dioxide-based urethane acrylate composites sealer incorporated with silver nanoparticles (AgNPs) and/or zinc oxide nanoparticles (ZnONPs) revealed that urethane-based resins can be created from isocyanate chemistry and raw materials with active hydrogens. Nanoscale silicate platelets (NSP) were used as carriers for nanoparticles. The incorporation of silver nanoparticles and zinc oxide nanoparticles increased antimicrobial activity by significantly reducing glucose uptake and obstructing the synthesis of adenosine triphosphate (ATP) for microbial growth. In the experiment, a biostable and non-toxic sealer was obtained, characterized not only by excellent resistance to hydrolysis and good mechanical properties but also, thanks to the addition of nanoparticles, excellent antimicrobial activities against *E. faecalis* [149].

Silver has a wide application in the modification of sealers while, when reduced to the nano-scale, it shows an increased surface area and manifests a strong antipathogenic activity at minor concentrations [150]. Silver has a broad-spectrum activity against bacteria with relatively low cytotoxicity [151]. The physicochemical analysis of silver nanomolecules (AgNPs) was also prepared by A. Teixeira et al. The mechanism of antimicrobial action was through their synergy with the thiol groups of enzymes in the metabolism that lead to bacteria death. The investigation was conducted against *E. faecalis*, *P. aeruginosa* and *E. coli*. In the experiment silver vanadate decorated with AgNPs (AgVO_3_) not only prevented agglomeration but also ensured thermodynamical stabilization of silver nanoparticles. The nanoparticles were mixed with a resin-based sealer (AH26 and AHPlus), zinc-oxide-eugenol-based sealer Endofill^®^ (PD Swiss, Vevey, Switzerland) and calcium hydroxide-based sealer (Sealer 26 and Sealapex. The study demonstrated that adding AgVO_3_ improved the antimicrobial properties of the endodontic sealers against *E. faecalis*, *P. aeruginosa* and *E. coli*. and could inhibit the growth of the tested microorganisms. Furthermore, the results showed that the use of nanoparticles did not deteriorate physical properties. The flow of new sealers was sufficient to fill the accessory canals and voids between the gutta-percha cones. They also exhibit radiopacity values higher than bone or dentin. In conclusion, this kind of materials may be advantageous during root canal obturation [152].

B.H. Baras et al. analyzed incorporating nanoparticles of silver (NAg) and nanoparticles of amorphous calcium phosphate (NACP). Methacrylate-resin dual-cured root canal sealer contained 5% DMAHDM to investigate the effects on the physical, anti-biofilm effect measured against *E. faecalis*, remineralizing ions, and hardness of human dentin. The dentin infection model was used. The researchers applied a 0.15% concentration of silver nanoparticles (AgNPs) and 10%, 20% and 30% of amorphous calcium phosphate nanoparticles (NACP). The novel therapeutic root canal sealer exhibited antimicrobial properties against *E. faecalis* in impregnated dentin blocks, related to detaching silver nanoparticles; through the antibacterial contact-killing properties of DMAHDM, it also reduced the biological membrane by almost 3 logs when comparing to the control group.

Further, it has shown the ability to raise the pH, remineralize and protect hard dental tissue against potential root fracture in conjunction the transport of calcium and phosphorus ions. It regenerated dentin minerals lost during irrigation with EDTA or NaOCl and increased dentin hardness. Moreover, the present study demonstrated that addition of DMAHDM, NAg, and NACP did not negatively affect the flow properties and film thickness [153]. To sum up, new sealers with dimethylaminohexadecyl methacrylate, nano-silver and nano-calcium phosphate may influence the killing of pathogens inside canals and increase mechanical properties of dentin after irrigation with sodium hypochlorite (NaOCl) and ethylenediaminetetraacetic acid (EDTA). This study stated that this material may increase safety and reduce complications during root canal treatment.

Similar conclusions were presented by Seung et al. The addition of dimethylaminododecyl methacrylate (DMAHDM) with nanosilver (NAg) into AH Plus™ was prepared. The authors revealed a significant reduction in bacteria on day one when the substance was added to the sealer, however, there was no difference between them at days 7 and 14. The highest concentrations of DMAHDM and NAg that did not affect the physicochemical parameters (setting time, flowability, solubility, and dimensional change) of AH Plus were determined. Incorporation of 2.5% DMAHDM and 0.15% NAg decreased the flowability of mAH Plus but it was still within the American National Standards Institute/American Dental Association specifications. In the results, no significant differences in setting time, solubility, or dimensional change were noticed.

### 3.3. Nano-Size Related Drug Delivery Applications in Endodontics

In recent years, nanotechnology has been developing rapidly in various fields of medicine. The development of nanotechnology is also taking place in medicine, including dentistry. Thanks to the variety of liposomes, micelles, polymer-based nanoparticles (NPs), nano-emulsions, nanogels, inorganic NPs, the nano-drug delivery system can have so many applications [154]. The small size of the drugs allows for their more precise application, which is also associated with their lower cytotoxicity. Drug delivery systems can be divided into nano drugs as their own carriers, nano drugs with nano carriers, nano drugs with other carriers or drugs with nanocarriers [155]. Nanofibers are one of the examples of drug carriers in endodontics. Thanks to their biocompatibility and resemblance to lost tissues and bactericidal properties, they can be an alternative to commonly used agents. In the case of pulp revascularization, various forms of nanoparticles (nanotubes, nanofibers, matrices) are used. The latter are classified into those compounded with polyacids (lactic, glycolic and caprolactone).

One of the many examples of nanomaterials’ utility in pastes is triple antibiotic paste (ciprofloxacin, metronidazole, and minocycline), with prophylene glycol as the carrier. Due to discoloration caused by minocycline, other antibiotics, such as ampicillin or clindamycin, are used [156]. J. Ribeiro et al. [157], in addition to lower discoloration and slightly lower bactericidal properties, noted the longer release of antibiotics from nanofibers. This may result in the demonstrated lower toxicity. In addition to revascularization, nanofibers have applications in other fields as well. M. Bottino et al. [158] mentions the application of growth factors in repair process in periodontal diseases. Chitosan is a natural polymer that is obtained from shells of crustaceans. It is characterized by such features as biocompatibility, biodegradability or lack of cytotoxicity. Thanks to the antimicrobial, anti-tartar and anti-plaque properties, nanochitosan is suitable for modifying the glass ionomer by increasing, depending on the concentration, the release of fluorine ions and increasing the strength of the material [159]. Chitosan also reacts with mucin, thereby improving the transport between the environment and the epithelium, due to the charge density associated with the deacetylation of the particles. Undoubtedly chitosan, thanks to its attributes, is applicable in variety of medicine areas [160]. 

Graphene is also one of the materials that is worth mentioning. It was only discovered in the current millennium and for this reason scientists still have many properties and applications to cover. This is due to its versatile properties (e.g., biocompatibility, microbicide, increased material strength, increased tissue regeneration), which are particularly valued in dentistry [161]. Graphene also has properties that favor the regeneration of tissues of the entire endodontium, which may be used in the future in endodontics [162]. Due to their properties, graphene and its compounds are suitable as a drug delivery carrier and as a cancer-fighting agent. In endodontics, doctors also deal with neoplasms, which are one of the greatest challenges of modern medicine. The results of the research are promising, but many things are not fully researched yet. It is definitely a substance that will play a large role in drug delivery systems in the future [163].

### 3.4. Root Repair Materials

Nanotechnology has also revolutionized the way of producing root repair materials. Nanometric particles significantly improved the physical parameters of the material, positively affecting treatment results. Obtaining nanoscale particles enables the design of materials with precise and ultra-fine architecture, which significantly improves the filling properties of the material. Nanoparticles are able to more accurately fit into the complex shape of the tooth canal. Improving the adhesion and tightness of the material used increases the percentage of successful root canal treatment [164,165].

Root repair materials, due to their biocompatibility, bioactivity and a number of physical features, are used in a variety of dental procedures, for example in endodontics. Mineral trioxide aggregate (MTA) is the first successful material used in treatment. You can use it in many situations in endodontics, such as perforations, root canal treatment, surgical treatment of apex resection. It can also be used for canal filling [166].

As described in a study by Mohammed S Alenazy et al. [167], the material of choice for backfilling is MTA, but it has long processing and setting times. To overcome these negative properties, studies used MTA in nanomodified particles for improved physicochemical properties. They discovered that when MTA was used in the form of nanoparticles, the surface area was increased for the powder in contact with tooth tissues, which makes setting time shorter and growth the hardness of the material. This change apparently helped MTA solidify faster. Unfortunately, the use of nanomaterials was associated with the deterioration in the required hardness after full setting. Additionally, despite its great recognition, MTA is not without its disadvantages—mainly in terms of biocompatibility, strength, setting time, and biological and physical properties. This limitation seems to be another challenge for nanomaterials in root repair materials.

#### Polymer Nanocomposites (PCN)

The term is used for all polymer materials that contain nanoparticles (for example, aluminum or carbon nanotubes). In these materials, the surface area to volume ratio is higher, which ensures better thermal and mechanical properties even with a low percent level of filler content in the material (up to 5%) [168].

The research also shows an upgrade in the physical attributes of materials related to classic materials—better heat resistance, dimensional stability, reduced electrical conductivity, and also the ability to release drugs [169,170,171,172]. There were also studies on the cytotoxicity of new polymer nanocomposites; no significant differences were found in the level of cytotoxicity of the material compared to classic materials [173]. Recent studies of two such nanocomposites (NERP1 and NERP2) have shown that NERP1 material significantly reduces micro-leakage around the tip compared to traditional material [174].

All this information suggests that the modification of traditional materials with nanoparticles may be the gold standard in the near future.

### 3.5. Nanoparticle-Based Disinfection in Endodontics

One of the most important elements of root canal treatment is cana disinfection by irrigation. Irrigation allows us to remove the infected tissues inaccessible to mechanical treatment alone. Theoretically, the irrigation fluid is able to reach all areas of the canals, removing pathological tissues from them, without damaging healthy tooth tissues. In modern times, using nanoparticles to irrigate the teeth canals has attracted attention. Silver nanoparticles are one of the materials used for this purpose. They have antibacterial and antifungal properties [175]. However, they are also likely to have inflammatory, oxidative, genomic and cytotoxic effects [176]. Researchers are looking for ways to use silver nanoparticles safely. An example is the use of antimicrobial photodynamic therapy based on nanoparticles [177]. In the work of Pagonis, the effect of polylactoglycolic acid (PLGA) nanoparticles together with the use of light was investigated (in vitro study). The study showed promising results—it reduced the number of *E. faecalis* bacteria in bacterial colonies.

Other studies confirming the effectiveness of nanoparticles against *E. faecalis* too [107,178]. The level of their effectiveness, however, is slightly lower than that of sodium hypochlorite. Interesting research has been carried out at Shiraz University of Medical Sciences [179], which presented that the final rinsing of tooth canals by the solution containing nanomaterials gives better fracture resistance compared to the use of NaOCl.

The effectiveness of nanoparticles is mainly provided by factors such as contact duration and particle size. Nanoparticles can reach places where standard formulations may have difficulties. These issues, however, require further research [180].

## 4. Nanomaterials in Endodontic Instruments and Their Effects

Nickel-titanium (NiTi) endodontic rotary files are common instruments in dentistry today. Most of those available on the market are clockwise rotating instruments; only a few are counterclockwise-rotated with reciprocation movement, e.g., Reciproc^®^ (VDW, München, Germany), Wave One^®^ (Dentsply Sirona, Charlotte, USA). Practically all of them have a non-cutting tip, except Mtwo Retritment^®^ (VDW, München, Germany), allowing one to file following the axis of the root canal or along the glide path. NiTi rotary files have different cross-sectional shapes which give them special properties. The smaller the cross-sectional area, the more flexible the tool and when the more acute angle at the edge, the cutting is better. Nitinol occurs in two crystalline phases (Figure 4) called martensite and austenite, and sometimes appears intermediate phase—R phase (e.g., Twisted Files). Their transformations (Figure 5) give particular properties, such as shape memory and superelasticity.

The characteristic of martensite (low-temperature phase) is a crystal structure with lower symmetry, such as tetragonal, rhombohedral, orthorhombic, monoclinic, or triclinic structure. Martensitic transformations are potentially reversible and take place without diffusion or plasticity. It happens form solid to solid phase (in crystalline structure) under the influence of stress or when temperature is change. Files in this phase are very flexible, not very springy, and resistant to cyclic material fatigue but less resistant to deformations resulting from screwing in. The temperature at which martensite changes into austenite is similar to human body temperature, therefore in the case of extremely curved root canals, it is advisable to cool the tool to make it more flexible. The characteristics of another phase, austenite (high-temperature phase) is very symmetrical crystal structure. Files in this phase are more elastic, stiff, resistant to deformation as a result of screwing in but less resistant to cyclic material fatigue [181].

The new Self-adjusting File (SAF) technology (Figure 6) is kind of compressible, resilient NiTi net, and without central core. The SAF technology allows for the flow of irrigant, effective disinfection and adapts for any shape of root canals, e.g., oval canals. It removes as minimal layer of dentin from around of canal as it is needed and help avoid removal of healthy dentin and due to this it does not cause micro-cracks in root dentin [182].

Another noteworthy NiTi rotary system is XP-Endo Finisher (Figure 7) whose revolutionary design enhances its ability to clean hard to reach places. It is characterized by extreme expansion capacity optimization up to 100× its core size, good adaptation for all root canals shapes, resistance to cyclic fatigue, flexibility and perfect removal of debris without removal significant amounts of good tissues.

These types of alloys have beneficial properties such as high corrosion resistance and superelasticity due to this good shape memory [182,183,184]. Cobalt coatings of the NiTi file with impregnated fullerene-like WS2 nanoparticles significantly improve the fatigue resistance and breakage time.

## 5. Nanoapplications for Repair and Pulp Regeneration

Teeth with degenerated and necrotic pulp are routinely saved with the use of root canal therapy. Nowadays we have excellent techniques of treatment, but in some cases pulp regeneration will be the ideal form of therapy, for instance when we can replace necrotic with healthy tissues, we do not weaken teeth tissues but revitalize them. In their study, Fioretti et al. [185] showed that a-MSH (melanocortin peptides) possess anti-inflammatory properties and also help the proliferation of pulpal fibroblasts and they reported it as a new active biomaterial for endodontic regeneration.

Nowadays, the most important concern in dentistry is to recover tissues and avoid endodontic treatment when it is possible. Treatment of reversible pulpopathies is based on regeneration through indirect (Figure 8) or direct pulp capping (Figure 9). In direct pulp capping, the protective dressing is placed directly over an exposed pulp and in indirect pulp capping, the protective dressing is put on the top of a thin layer of softened dentin, because removing it would expose the pulp. The removal of irritants and materials, and the occurrence of defense and regenerative mechanisms promotes pulp healing and the generation of reparative dentin (sclerotic dentin). Treatment prognosis is better when the exposed pulp does not become infected or inflamed. When dental caries develop and become deeper and closer to the pulp, risk of pulpal inflammatory is higher due to penetration of microorganisms and their toxic products. When the caries are removed at an early stage, risk of the damage, alterations in capillary filtration, the pulpal inflammations and the post-operative hypersensitivity is much lower, despite the significant obstacle, because pulp possesses a considerable ability to repair itself [2].

In endodontics, repair and regeneration have a close relationship with the embryonic stem cells growth to maintaining and restoring original structures and functions.

Characteristics of human pulp are terminal micro-vascular supplies, few anastomoses, and a generally large volume of tissue with generally little vascular supply, making it difficult to interfere with the pulp without causing inflammation [186].

It is confirmed, that the regeneration and removal of inflammation is related to many factors [187], e.g., components of stem and progenitor cell proliferation, differentiation development, scaffold types, the regeneration of neural and vascular tissues and mechanisms of signaling and the proteins that are involved in the process of signaling.

An odontoblast is a cell that takes part in a process of dentin formation called dentinogenesis. These cells form the outer surface of the dental part and adjoin with dentin. An odontoblast is a non-dividing cell that functions over the lifetime of dental pulp. An odontoblast-like cell appears when it is replaced by a cell. Further on, when an infection or an injury arises, it gives rise to a dentin-like structure. Secondary dentin forms during the lifetime of a tooth as a physiological process. It derives from activities of the original odontoblastic layer by the same process as the primary dentin. The vitality of the pulp is necessary for the proper functioning of the tooth. Many researchers are studying the impact of odontotropic properties on pulp regeneration and how to avoid the removal of all the pulp following irreversible pulpitis [2]. Some studies show that nanotechnology would provide suitable solutions for approaches to pulp tissue conservation and regeneration.

A recent study by Fioretti et al. [185] tested the toxicity of nanostructured assemblies on dental pulp tissues, and the anti-inflammatory properties of alpha-melanocyte-stimulating hormone (MSH) were demonstrated. Their group have shown that the combination of substances with Poly-Glutamic Acid (PGA) with the incorporation of an anti-inflammatory hormone (melanocortin, a-MSH) into the multilayered Poly-L-Lysine (PLL)/PGA films increases the anti-inflammatory reaction of pulp fibroblasts (causing the proliferation of fibroblast cells) and macrophages stimulated by LPS (Lipo-Polysaccharides) [188]. These nanostructured assemblies are a reservoir of the anti-inflammatory peptide and promote the adhesion and proliferation of pulp fibroblasts on the biomaterial. 

Smith IO et al. [189] and other researchers developed and examined nanostructured polymer scaffolds. They proved that the structural features of tissue engineering scaffolds affect cell response and must support cell adhesion, proliferation and differentiation. The test focused on nanofibrous (NF) scaffolds with combinations of components, which was similar to a synthetic extracellular matrix (ECM) interacting with cells before forming new tissue. Their group has developed biodegradable polymer arising in TIPS process to form NF with nanofibers with the same size and diameter as the collagen fibers found in the ECM (diameter 50–500 nm). This group has also grown apatite crystals onto biodegradable polymer scaffolds (by SBF), in which they changed the quantity and schedule of these crystals. Researchers are working on the schedule of these crystals throughout three-dimensional scaffolds (in both nano-fibrous and composite), to improve the ability of the cell to adhere, proliferate, and differentiate. It is also possible through growth and differentiation factors.

The production of an engineered replica of the naturally occurring ECM can promote the development of new tissue which is important for tissue repair and regeneration. Yang et al. [190] studied in vitro and in vivo behavior of dental pulp stem cells (DPSCs) on different scaffolds such as poly(epsilon-caprolactone) (PCL)/gelatin scaffolds with or without the addition of nano-hydroxyapatite (nHA). In the in vitro evaluation, DNA content, alkaline phosphatase (ALP) activity and osteocalcin (OC) measurement showed that the scaffolds supported DPSC adhesion, proliferation, and odontoblastic differentiation. In conclusion, the incorporation of nHA in nanofibers indeed enhanced DPSCs differentiation towards an odontoblast-like phenotype in in vitro and in vivo study models.

One study assessed the differentiation of human odontogenic DPSCs on NF poly L-lactic acid (PLLA) scaffolds. Wang J. et al. [191] show that odontogenic differentiation of DPSCs can be achieved on NF-PLLA scaffolds and the combination of BMP-7 and DXM induced the odontogenic differentiation more effectively than DXM alone. The NF-PLLA scaffold and the combined odontogenic inductive factors provide an excellent environment for DPSCs to regenerate dental pulp and dentin.

Gupte MJ et al. [192] with his group used highly permeable scaffolds of NF-PLLA (mimicked collagen type-I fibers) with and without the usage of the growth factors Bone Morphogenic Protein-7 (BMP-7) and dexamethasone (DXM) medium. It presented that mixing of the growth factor BMP-7 and DXM stimulated the differentiation of odontogenic DPSCs more effectively than DXM alone. The obtained environment was excellent support for DPSCs in regenerating dental pulp, dentin, and enamel.

## 6. Discussion

Although nanotechnology appears to introduce ground breaking techniques and devices in the dental field, it also presents concerns. These include economical nanorobot mass production, ethical issues and human safety, biocompatibility issues and expertise in precise positioning and technique. Nanotechnology is expected to change healthcare fundamentally by providing novel methods for disease diagnosis and prevention, therapy selection tailored to the patient’s profile, drug delivery and gene therapy.

The nanotechnology sector in dentistry is developing very dynamically. Reduction of the size of particles to nanoparticles may lead to several new physicochemical and biological properties, and therefore many potential applications, not only in dentistry. Expectations are high, in particular for nanotechnology in endodontics, and research shows that its clinical application is very promising.

Nanomaterials can be used in a variety of ways in endodontics, as Table 1 presents. Their great variety allows for interesting applications in several fields of dentistry, which clearly reflects their potential in dental clinical use [5,193,194,195,196]. The greatest influence on clinical outcome is exerted by their unique size-related characteristics [18].

It is necessary to integrate knowledge based on biology, classical physics as well as quantum mechanics to apply nanotechnology and achieve better results in endodontic treatment. Nanoparticles can be applied in various aspects of endodontics and can significantly affect the physical and mechanical properties of materials they are mixed with.

### Challenges and Advantages of Endodontic Nanotechnology

Endodontics still remains a challenging specialization in dentistry, due to the complicated and diverse anatomy, as well as poor accessibility of the object of its interest [208].

Nanomaterials may prove to be the turning point in the effective endodontic treatment. The current literature suggests that nanoparticles have distinct advantages due to other substances including medications, bioactive compounds and photosynthesizers, and better anti-pathogenic properties [126]. Moreover, they can be used in vital pulp therapies, irrigation of canal disinfectants, intracanal medicaments, obturation materials and regenerative endodontic materials. For instance, in case of dental pulp regeneration, Bellamy et al. [209] in their experiment presented the potential of a carboxymethyl chitosan-based scaffold with growth factor releasing nanoparticles to induce migration and differentiation of stem cells from the human apical papilla.

Nanoparticles commonly have new properties not found in their macro counterparts. The advantages of these materials are antimicrobial activity, mechanical reinforcement, aesthetics, and therapeutic effects [210].

The search for perfect materials led to in vivo and in vitro experiments in which the incorporation of nanoparticles improved physicochemical and bioactive properties. The oral cavity is a place of pathogens that can cause caries, pulpitis, gingivitis or periodontitis, and in the case of teeth after endodontic therapy, their reinfection and the occurrence of complications.

The most common reasons why root canal treatment fails is the presence of pathological organisms inside the root, hence the use of dental materials modified/functionalized with nanoparticles with antibacterial properties along with chitosan nanoparticles, hydroxyapatite nanoparticles, metal nanoparticles, polymeric nanoparticles, inorganic bioactive nanoparticles and calcium silicate nanoparticles [124]. Most nanomaterials have a similar mechanism of action based on penetrating biofilms. What is important is that they interact with bacterial cell walls and disrupt cellular processes which ultimately lead to cell death.

One of the great benefits of nanoparticles is the capacity to reach the lateral canals and apical ramifications of the canal’s apical third and fight pathogens not only by eliminating their planktonic form but also by disrupting biofilm matrix.

Silver nanoparticles are preferred above all others metal nanoparticles; their advantage is the fact that they exhibit antimicrobial and antifungal activity. Their use in mixture with antibiotics enhances their antibacterial action, especially against strains resistant to traditional antibiotics.

In addition, a small amount of silver is required to effectively stimulate the action mechanisms of antibiotics, and therefore the cytotoxicity of AgNP on human cells is negligible [204]. The synergistic and multimode antibacterial actions substantially reduce the extensive use of antibiotics, and therefore minimize the potential of antibiotic resistance and toxicity to the neighboring tissues.

Different metal oxide nanoparticles, such as titanium dioxide, magnesium oxide and iron oxide, have also proven to be effective as bactericidal agents. There has also been enthusiasm over the use of iron oxide nanoparticles due to their strong anti-biofilm properties, high biocompatibility and therefore lack of harmful effects on oral tissues. These nanoparticles have an intrinsic peroxidase-like activity which enables them to catalyze hydrogen peroxide. Due to the production of free radicals, they exhibit bactericidal anti-biofilm activity.

An important advantage of nanomaterials used in endodontics is their improved depth of penetration into dentinal tubules, as in the case of calcium hydroxide nanoparticles. Thanks to the increased contact surface with pathogens and better solubility, they show strong antimicrobial activity. In addition, their advantage over conventional calcium hydroxide used in intracanal medicaments is the decrease of dentin microhardness. Research shows that most dental materials used for irrigation and obturation, and intracanal medicaments enriched with nanoparticles, can destroy not only planktonic organisms but also bacterial extracellular polymer substance (EPS) and can also eradicate biofilm. Examples of nanoparticles can be chitosan and zinc oxide which show a broad antimicrobial spectrum. A. Shrestha at el. proved that they continued to be active even after aging for 90 days [211].

The antipathogenic effect, especially against *E. faecalis*, is improved by combining nanoparticles with photodynamic therapy. Rose bengal-conjugated chitosan was able not only to inactivate endotoxins but also completely eliminate the microorganisms after 24 h [205], The use of photodynamic therapy with nanoparticles may also stabilize the dentin matrix and improve the physical properties of dentin by inhibiting collagenolytic activity. It increases the risk of fracture. Improved antimicrobial activity can also be obtained through the synergistic action of compounds containing nanoparticles.

One of the many advantages of using nanoparticles in endodontic treatment is the influence on the structure of root dentin. Research shows that the final irrigation of root canals with nanoparticle solutions including silver nanoparticle (SNP), titanium dioxide nanoparticles (TNP) and zinc oxide nanoparticles (ZNP) increased the fracture resistance of root canal-treated teeth. This is possibly related to improved wettability and increased dentin surface energy when irrigated with nanoparticle solutions or better contact of the nanomaterial with dentin and, subsequently, the better infiltration of particles into collagen fibers. However, further research is required. The studies by Jowkar Z. et al. [179] demonstrated comparison of fluids to canal final irrigation. The results obtained almost 400 N higher mean fracture resistance of teeth after endodontic treatment with ZnONPs-based solution in comparison with NaOCl [126]. The strengthening of root dentin and higher remineralization potential was also obtained by adding nanoparticles to obturation materials. Adding QPEI nanoparticles into AH PlusTM, an epoxy resin-based sealer, and Pulp Canal SealerTM, a zinc oxide eugenol-based sealer, was investigated to stimulate osteoblastand osteoclast proliferation [212] and promote the formation of new hard tissue. Antibacterial activity and remineralization potential were also observed in the combination of AgNPs and DMAHDM [213]. Another example showed in a study by Hashmi A. et al. demonstrates that conditioning of dentin with chitosan-hydroxyapatite precursor (C-HA) nano complex before the use of tricalcium silicate material was shown to result in a chemically modified dentin and improved sealer penetration into dentinal tubules [214].

The implementation of new technologies carries risks not only in terms of potential toxicity to human tissues, but also is a global public health challenge, hence the need to thoroughly discuss risk assessment issues at each stage of the life cycle of nanotechnology-based products as well as establish detailed protocols and testing strategies for nanoparticles used in endodontic treatment. The challenge is the toxicity of nanoparticles and their different biological properties compared to their “macroscopic” counterparts [215] attributable to their unique chemical and physical properties. The problem may be not only their use but also their production, storage, transport, permanent disposal and recycling policies.

The resulting nanostructured materials often have significant limitations related to their biopharmaceutical and pharmacodynamic properties. When penetrating biological barriers, they can accumulate in non-target sites and undergo chronic bioaccumulation as a result of attaching to proteins, cell membranes and other biological structures. Due to their dimensions similar to biological molecules, they can be easily absorbed by different tissues and organs and can be accumulated in the liver, lungs, and the reticuloendothelial system [216]. The mentioned processes can cause damage to cells and tissues through increased oxidative stress, blocking ion channels, or mechanical damage to the cell membrane and other organelles [217]. These factors can also interfere with cell proliferation, resulting in cell death or uncontrolled growth.

Factors which influence the toxicity of nanomaterials are related to aggregation, geometry and surface charge, particle size, the material, concentration and duration of exposure [189]. The main challenge in working with nanoparticles is their tendency to agglomeration or aggregation due to the high proportion of surface atoms in relation to volume atoms; therefore, it can be difficult to maintain a stable compound structure and inertness in the oral environment. Moreover, limitations in endodontics may result from direct interaction with the surrounding environment of hard or soft tissues [218].

Another challenge for the use of nanomaterials in endodontics is related to their manufacturing. High production costs can be an obstacle to the mass production of nanoparticles in the medical industry. Nanomaterials are produced in most cases on a laboratory scale, which means high costs of their synthesis and a long time required to implement the process of their production. What is more, there are often problems related to both the repeatability and reproducibility of products with appropriate stability and dimensions not deviating from the assumed size.

## 7. Conclusions

The primary focus of endodontics is the elimination of biofilm-forming bacteria. The future of dentistry, especially endodontics, will change with nanotechnology which has an immense potential of exerting a profound influence on healthcare and human life.

Nanoparticles have many properties that are beneficial to modern dentistry. Thanks to their use, regenerative therapies can be more effective. Nanoparticles serve as carrier systems of active agents to stimulate stem cell proliferation, migration and differentiation, or as a scaffold constituting temporary structures that promote the growth and differentiation of stem cells. Another advantage of nanomaterials is their effect on the adhesion of obturation and restorative materials to root canal walls.

These facts prove that nanotechnology is already a field of great importance in the treatment of oral diseases and that its therapeutic application in root canal treatment will increase in the coming years. However, it should be noted that most of the research on dental nanomaterials is in vitro studies. There are few publications reporting the impact of nanomaterials in vivo, and in some cases the improvement resulting from the application of nanoparticles is minimal or ambiguous. The lack of reliable and long-term studies, both in the animal model and in clinical trials with patients, is a challenge and requires special care with their use. It is necessary to conduct further research following the high standards of evidence-based medicine (EBM), as well as careful follow-up after the end of treatment, to assess the long-term effects of nanoparticles in endodontic treatment, in particular their immunogenicity, biocompatibility and/or or biodegradability.

Nanodentistry is a rapidly growing field; the application of new molecular products and techniques can improve oral health conditions through better diagnosis and treatment of dental problems. Nanotechnology can break down barriers to conventional technology, however there are many ethical, social and legal issues that need to be resolved before being introduced to dentistry and medicine.

## Figures and Tables

**Figure 1 materials-14-05296-f001:**
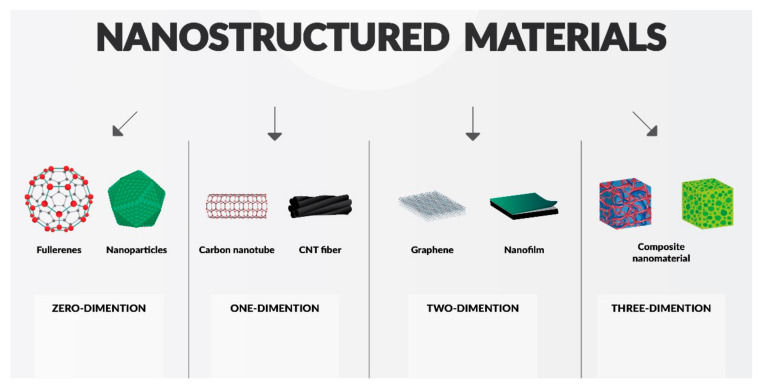
Graphical representation of the nanostructured materials dimensional division.

**Figure 2 materials-14-05296-f002:**
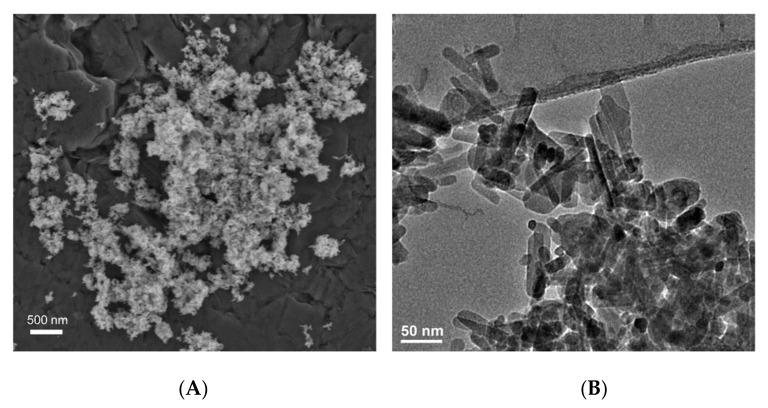
(**A**) SEM image of the nHAp; (**B**) TEM image for nHAp.

**Figure 3 materials-14-05296-f003:**
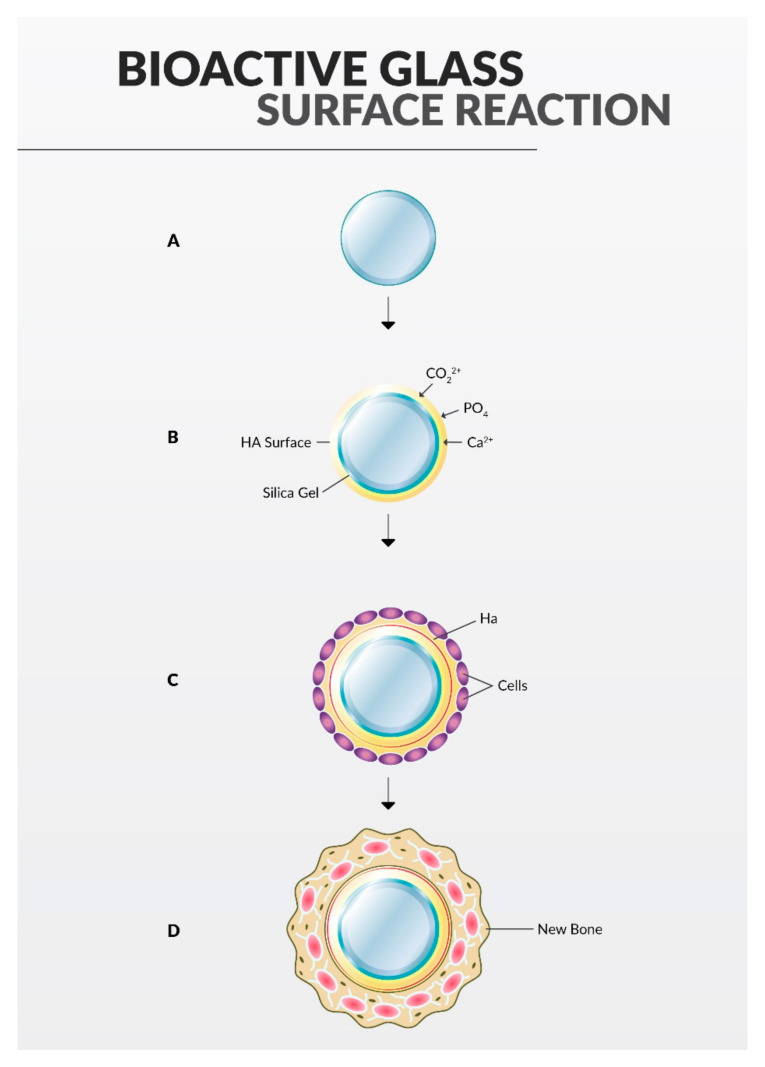
Formation of new bone tissue on bioactive glass. (**A**)—bioactive glass. (**B**)—adhesion of ions to the sillica surface, which results in a formation of bone-like HA. (**C**)—osteogenic cells cover the surface of the hydroxyapatite and create a coated bioactive glass. (**D**)—crystallization leads to the formation of a new bone tissue.

**Figure 4 materials-14-05296-f004:**
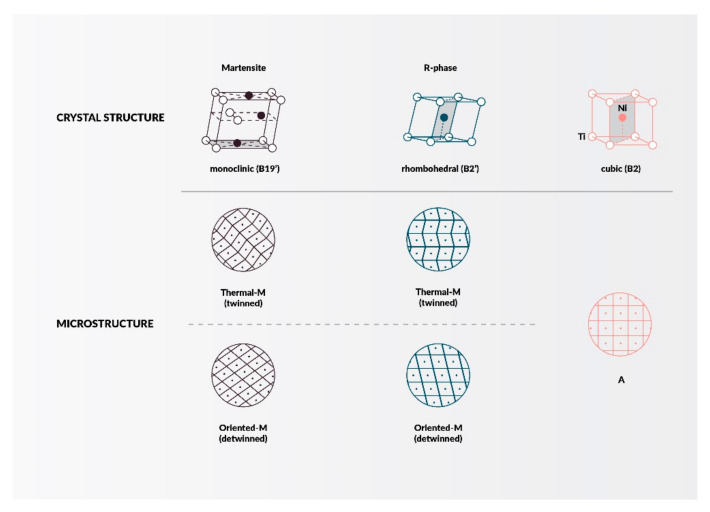
Different crystal structure phases and their microstructural characteristics.

**Figure 5 materials-14-05296-f005:**
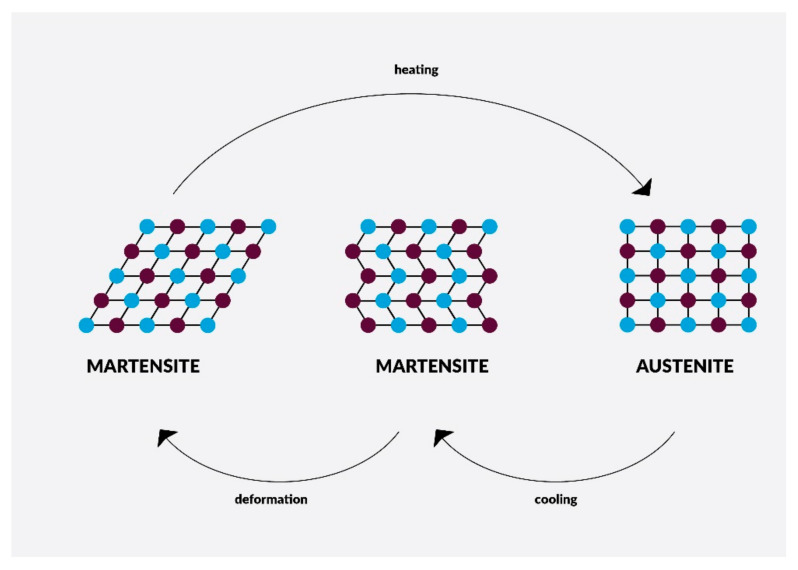
Graphic presentation of crystalline phases and their transformations.

**Figure 6 materials-14-05296-f006:**
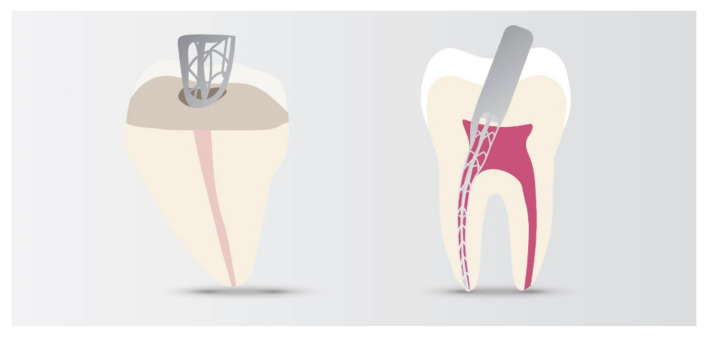
Self-adjusting File SAF technology visualization.

**Figure 7 materials-14-05296-f007:**
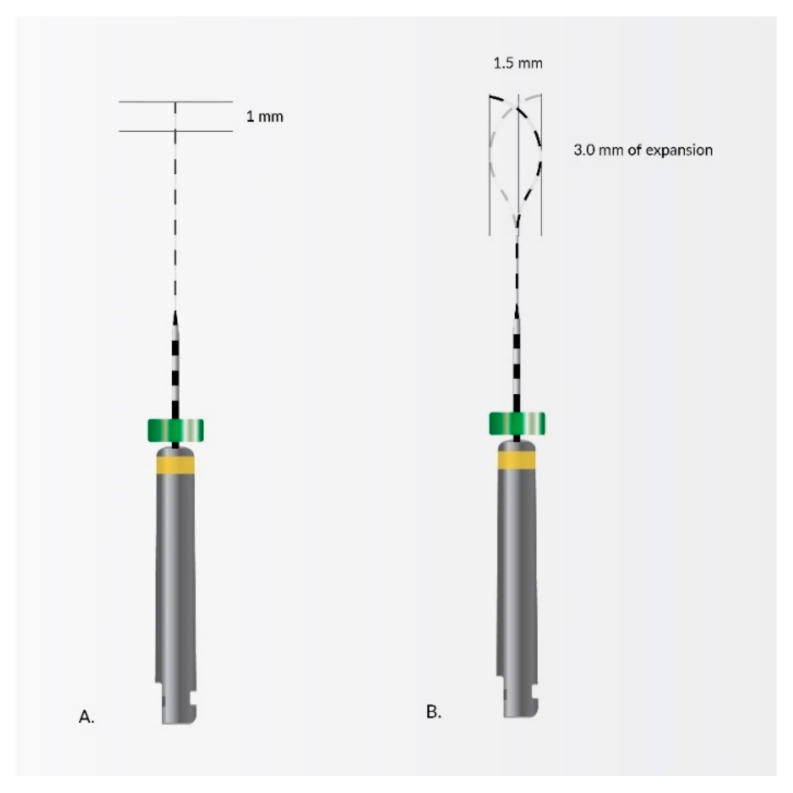
NiTi rotary system XP-Endo Finisher.

**Figure 8 materials-14-05296-f008:**
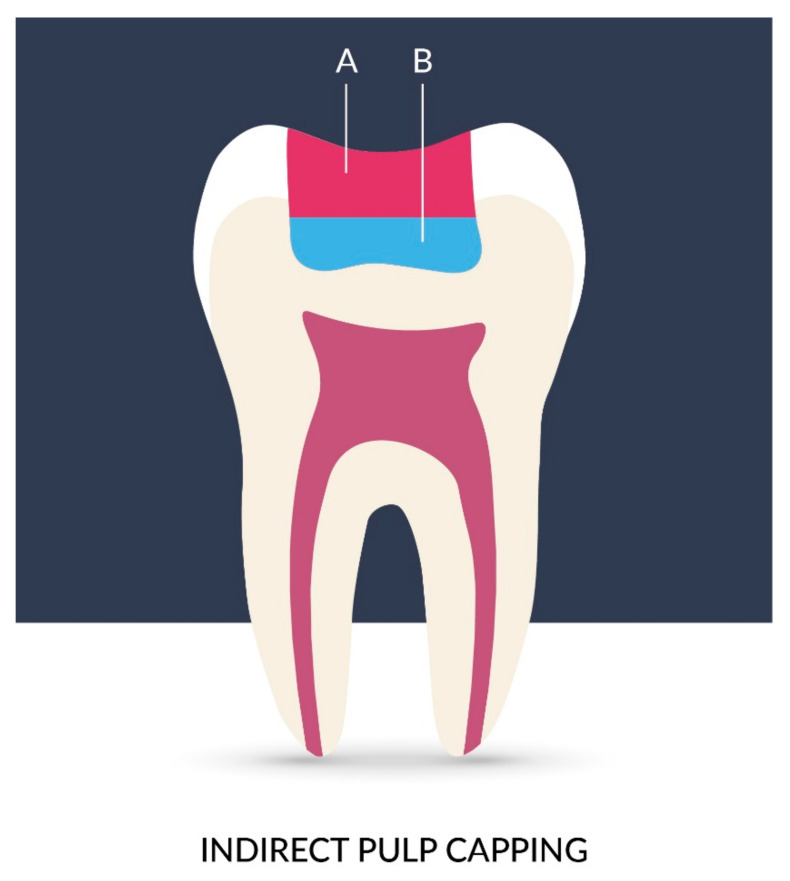
Graphical visualization of indirect pulp capping technique. (**A**)—restorative material; (**B**)—indirect pulp capping material.

**Figure 9 materials-14-05296-f009:**
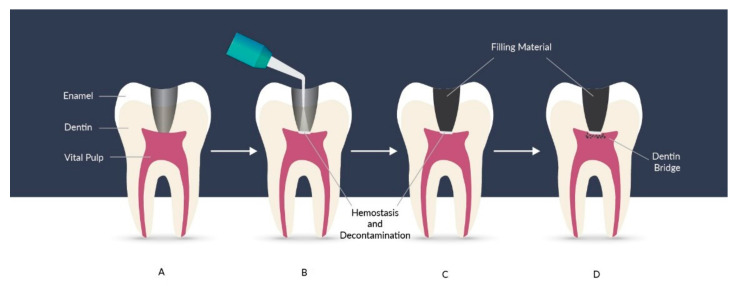
Treatment steps of direct pulp capping technique. (**A**) exposure of vital pulp. (**B**) Hemostasis and decontamination of the exposed pulp tissue with a use of laser. (**C**) Application of filling material after laser application and successful hemostasis. (**D**) Formation of dentin bridge.

**Table 1 materials-14-05296-t001:** Use of nanomaterials in endodontics.

Nanomaterial	Application	Method of Use	Details	Reference
Ca(OH)_2_	Sealing	Sealing Application of a sealer on a guttapercha point followed by canal obturation	Study focused on reduction of microleakage from the apical to coronal third by exchanging traditional ZnoE sealer with Ca(OH)_2_	[197]
nAg	Root restoration	Application into root cavities	Material has been mixed with MTA and placed in the cavity	[198]
Nano zinc oxide	Sealing	Canal obturation with a sealer	Zinc oxide nanoparticles mixed in a zinc-oxide eugenol cement	[199]
nAgVO_3_	Sealing	Canal obturation with a sealer enriched with a nanomaterial	AH Plus endodontic, Sealer 26, Endomethasone N sealers were incorporated with AgVO_3_	[200]
Nanodiamond	Sealing, Antimicrobial action	Canal obturation with a nanodiamond guttapercha	Nanodiamond-amoxicillin conjugates are conjugated and incorporated into a gutta-percha point	[137]
Chitosan nanoparticles	Sealing	Nanomaterial added to the sealer	Chitosan nanoparticles added to the epoxy resin sealer elevate the apical sealing ability of root canal obturation	[201]
Chitosan nanoparticles	antimicrobial	Nanoparticles as a part of irrigating solution	Several irrigants were compared, including one with chitosan nanoparticles, according to the study, it can be a useful alternative to EDTA irrigants	[202]
Nano-sized Liposomes	Drug delivery	Nano-sized biomaterial used for application of drugs	80–100 nm sized liposomes with incorporated drugs specifically used for resistant bacterial biofilm that causes persistent endodontic and periodontic disease	[203]
nAg	antimicrobial	Nanoparticles used as an antimicrobial material	Different concentrations of nAg particles were tested for their antimicrobial effect on *E. coli*, *S. aureus* and *P. aeruginosa*	[204]
Chitosan nanoparticles	Photodynamic therapy	Nanoparticles have been a photosensitizer increased its efficacy	Application of chitosan nanoparticles into a photosensitizer increased activity of the material. Their higher surface area resulted in increased binding of the photosensitizer to the particles.	[205]
Quaternary Ammonium Polyethyleneimine nanoparticles (QPEI)	Antimicrobial resins	Nanoparticles were mixed with dental resins	QPEI have been chemically incorporated into several polymeric resin matrices of dental materials, which resulted in a long-lasting antimicrobial effect.	[206]
Nanohydroxyapatite	Regeneration therapy	Dental pulp stem cells have been seeded on the nanocomposite scaffolds	Nanohydroxyapatite scaffolds elevated in vivo and in vitro odontogenic differentiation of the Dental pulp stem cells	[190]
Nanofibrous poly (L-lactic acid) scaffolds	Regeneration therapy	Dental pulp stem cells have been seeded on the nanomaterial scaffold	Nanofibrous poly (L-lactic acid) scaffold increased the odontogenic differentiation of dental pulp stem cells	[191]
Exosomes	Regeneration therapy	Exosomes derived from dental pulp stem cells were tested for their influence for endothelial cells regeneration	Exosomes derived from dental pulp stem cells highly enhanced the proliferation, angiogenesis and migration of endothelial cells in vitro and accelerated in vivo cutaneous wound healing	[100]
Graphene oxide	Antimicrobial	Silver nanoparticles were synthesized on graphene oxide particles	On an ex vivo model of an infected tooth, silver-graphene oxide material has successfully achieved and enhanced antimicrobial activity	[207]

## Data Availability

Not applicable.

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
