# Peer review of "Nanomaterials Application in Endodontics"

_materials, 2021, doi:10.3390/ma14185296_

Round 1

Reviewer 1 Report

The field of endodontics has been the focus of many recent advances in material sciences and equipment. The continuous evolution in endodontic materials is very promising, considering the dependence of this dental field on materials with ideal properties to achieve successful outcomes.

The manuscript approach is the complex one, but with some disadvantages regarding a better understanding for all readers.
1. The introduction it’s summary, I suggest more information about the basic knowledge in endodontics. Also it is not very clear the purpose of this review, I suggest it be added at the end of introduction.

  1. I understand that the review is long, but at point 2. Characterization of nanomaterials should be added to some images regarding the characterisation of studied materials (SEM, TEM).
  2. The strong point of this manuscript is the complexity approach of the nanomaterials and nanotechnologies used in endodontics.
  3. The weakness of the study is the exemplification, in images, of the characterisation of the materials and endodontic procedure.
  4. The conclusions must be reorganised, and a part of text which includes bibliography, must be added at the discussion section.

Author Response

Dear Reviewer,

We would like to express our sincerest gratitude to the Reviewers for their enormous efforts in criticizing the manuscript. All remarks have been included in the revised version of the manuscript.

Question 1: The introduction it’s summary, I suggest more information about the basic knowledge in endodontics. Also it is not very clear the purpose of this review, I suggest it be added at the end of introduction.

Answer: We would like to thank you for the comment. We have revised the introduction part according to the reviewer’s suggestions.

Question 2: I understand that the review is long, but at point 2. Characterization of nanomaterials should be added to some images regarding the characterisation of studied materials (SEM, TEM).

Answer: We would like to thank you for the comment. SEM and TEM images have been added to the review.

Question 3: The weakness of the study is the exemplification, in images, of the characterisation of the materials and endodontic procedure.

Answer: We would like to thank you for the comment. The figures have been changed in order to to be more readable and useful for the reader.

Question 4: The conclusions must be reorganised, and a part of text which includes bibliography, must be added at the discussion section.

Answer: We would like to thank you for the comment. The conclusions part has been reorganized, additionally, the text part with the bibliography has been moved to the discussion section as the reviewer suggested.

Reviewer 2 Report

The following issues need to be addressed carefully before this work can be accepted for publication.

  1. The Introduction section is oversimplified. The readers of the journal “Materials” may be not experts in dentistry. It is authors’ responsibility to provide the readers with the basic knowledge of endodontics, for example via a schematic illustration. Also, the three sentences describing nanomaterials are independent to each other and have weak connection to the context. The general functions and application potentials of nanomaterials in endodontics may be introduced. In addition, the authors should tell readers the main contents of this present review.
  2. Section 2 is entitled as “Characterization of nanomaterials, materials modification”. However, the context has little relation to “characterization” but is highly relevant to “classification”. In this regard, some important nanomaterials are missing, such as exosomes, graphene-based nanomaterials and polymeric nanoparticles.
  3. Drug delivery is one of the most important functions of nanomaterials, and the authors also mentioned it. Peng's group recently has summarized the potentials of various nanomaterials in drug delivery ( J Controlled Release, 2018, 286: 64-73). In addition, the authors discussed the antimicrobial activity of some types of nanomaterials. Recently, Peng's group has made great efforts to develop drug-free antibacterial nanomaterials (J Controlled Release, 2019, 307: 16-31). Still, not everything need be discussed, it is still a useful addition to the literature so as to highlight the great potentials of nanomaterials in drug delivery and antibacterial therapy that has been somewhat neglected.
  4. The figures may be re-designed so as to reduce the margins and enlarge the phrases.
  5. Section 7 should be integrated into Section 6 since they are tightly related.
  6. Table 1 is too simple and should be enriched with some other critical items of nanomaterials, such as size, zeta potential and/or loaded drugs or agents.
  7. The large surface area of nanomaterials leads to a high surface energy, which causes a high potential of nanomaterials to interact with biomacromolecules and form corona (Biomacromolecules, 2019, 20(4): 1789-1797). The formation of protein corona in oral cavity following the interaction of nanomaterials with the proteins in the mouth and its impacts on the use of nanomaterials in endodontics may be discussed.

Author Response

Dear Reviewer,

We would like to express our sincerest gratitude to the Reviewers for their enormous efforts in criticizing the manuscript. All remarks have been included in the revised version of the manuscript.

Question 1: The Introduction section is oversimplified. The readers of the journal “Materials” may be not experts in dentistry. It is authors’ responsibility to provide the readers with the basic knowledge of endodontics, for example via a schematic illustration. Also, the three sentences describing nanomaterials are independent to each other and have weak connection to the context. The general functions and application potentials of nanomaterials in endodontics may be introduced. In addition, the authors should tell readers the main contents of this present review.

Answer: We would like to thank you for the comment. The introduction part has been revised by authors according to the reviewers’ suggestions.

Question 2:  Section 2 is entitled as “Characterization of nanomaterials, materials modification”. However, the context has little relation to “characterization” but is highly relevant to “classification”. In this regard, some important nanomaterials are missing, such as exosomes, graphene-based nanomaterials and polymeric nanoparticles.

Answer: We would like to thank you for the comment. Section 2 has been revised and supplemented with additional nanomaterials according to reviewer’s suggestions.

Question 3: Drug delivery is one of the most important functions of nanomaterials, and the authors also mentioned it. Peng's group recently has summarized the potentials of various nanomaterials in drug delivery (J Controlled Release, 2018, 286: 64-73). In addition, the authors discussed the antimicrobial activity of some types of nanomaterials. Recently, Peng's group has made great efforts to develop drug-free antibacterial nanomaterials (J Controlled Release, 2019, 307: 16-31). Still, not everything need be discussed, it is still a useful addition to the literature so as to highlight the great potentials of nanomaterials in drug delivery and antibacterial therapy that has been somewhat neglected.

Answer: We would like to thank you for the comment. The drug delivery section has been revised by the authors and additional bibliography has been added.

Question 4: The figures may be re-designed so as to reduce the margins and enlarge the phrases.

Answer: We would like to thank you for the comment. The figures have been re-designed according to the reviewer’s suggestions.

Question 5: Section 7 should be integrated into Section 6 since they are tightly related.

Answer: We would like to thank you for the comment. Mentioned sections have been integrated.

Question 6:  Table 1 is too simple and should be enriched with some other critical items of nanomaterials, such as size, zeta potential and/or loaded drugs or agents.

Answer: We would like to thank you for the comment. Table 1 has been extended with additional information and nanomaterials.

Question 7:  The large surface area of nanomaterials leads to a high surface energy, which causes a high potential of nanomaterials to interact with biomacromolecules and form corona (Biomacromolecules, 2019, 20(4): 1789-1797). The formation of protein corona in oral cavity following the interaction of nanomaterials with the proteins in the mouth and its impacts on the use of nanomaterials in endodontics may be discussed.

Answer: We would like to thank you for the comment. The following scientific work has been cited and the  nanomaterials’ corona phenomenon has been mentioned in the review.

Reviewer 3 Report

Recently, research in the field of biomaterials and nanotechnology is affecting also new dentistry materials development. The major challenge in this field is biofilm removal and infections preventions, so new nanosized materials are needed for these purposes. The paper collects the best currently available knowledge about nanomaterials used in endodontics and presents a future perspective of this research field.

The strength of the article is the clear writing and effective schematization, which makes this work suitable for offering as reading in Ph.D. journal clubs.

The weakness of the work is that the images presented are only drawings, although these drawings are useful and clear. There are no images taken during the performance of endodontic procedures, or electron microscopy (TEM and SEM) or atomic force microscopy images, in section 2 on nanomaterial characterization.

In My opinion, the paper might be published with no changes, but if it was enriched with images it would be more complete. Regarding the English, it looks to be well-written, but I'm not an English native speaker and I cannot make a judgment on this.

Author Response

Dear Reviewer,

We would like to express our sincerest gratitude to the Reviewers for their enormous efforts in criticizing the manuscript. All remarks have been included in the revised version of the manuscript.

Question 1: The weakness of the work is that the images presented are only drawings, although these drawings are useful and clear. There are no images taken during the performance of endodontic procedures, or electron microscopy (TEM and SEM) or atomic force microscopy images, in section 2 on nanomaterial characterization.

Answer: We would like to thank you for the comment. In order for the review to be more readable and understandable for the reader, the authors have extended the table with possible applications of nanomaterials in dentistry. Additionally, TEM and SEM images have been added to the review.